# Do Smaller Language Models Answer Contextualised Questions Through Memorisation Or Generalisation?

## Abstract

A distinction is often drawn between a model's ability to predict a label for an evaluation sample that is directly memorised from highly similar training samples versus an ability to predict the label via some method of generalisation. In the context of using Language Models for question-answering, discussion continues to occur as to the extent to which questions are answered through memorisation. We consider this issue for questions that would ideally be answered through reasoning over an associated context. We propose a method of identifying evaluation samples for which it is very unlikely our model would have memorised the answers. Our method is based on semantic similarity of input tokens and label tokens between training and evaluation samples. We show that our method offers advantages upon some prior approaches in that it is able to surface evaluation-train pairs that have overlap in either contiguous or discontiguous sequences of tokens. We use this method to identify unmemorisable subsets of our evaluation datasets. We train two Language Models in a multitask fashion whereby the second model differs from the first only in that it has two additional datasets added to the training regime that are designed to impart simple numerical reasoning strategies of a sort known to improve performance on some of our evaluation datasets but not on others. We then show that there is performance improvement between the two models on the unmemorisable subsets of the evaluation datasets that were expected to benefit from the additional training datasets. Specifically, performance on unmemorisable subsets of two of our evaluation datasets, DROP and ROPES significantly improves by 9.0%, and 25.7% respectively while other evaluation datasets have no significant change in performance.

## 1   Introduction

Memorisation has been described as the learning of a direct mapping between input features and particular outputs (Chatterjee, 2018; Elangovan et al., 2021; Schwarzschild et al., 2021; Lewis et al., 2021), in contrast with generalisation (Elangovan et al., 2021), or the application of a method for deriving the output (Schwarzschild et al., 2021). A number of studies have considered the impacts of memorisation from the perspective of the capacity of particular models to memorise pretraining data e.g. Carlini et al. (2023); Chowdhery et al. (2022) as well as through the lens of downstream evaluation dataset contamination e.g Brown et al. (2020); Sanh et al. (2021); Wei et al. (2021); Du et al. (2022); Chowdhery et al. (2022). A general finding has been that memorisation capacity scales with model parameter count, which implies that smaller models would suffer less from this problem. However observations from Lewis et al. (2021); Hartill et al. (2023) on the BART model (Lewis et al., 2020) suggest that undetected memorisation could effect smaller Language Models sufficiently so as to be an issue in interpreting results.

We consider the impact of memorisation on evaluation samples that should preferably involve reasoning from a question, over a provided context to an answer. Where the context is of a free-form nature we describe these as requiring reading comprehension (RC samples) and we denote samples where the context comprises multi-choice options as MC samples. We characterise an evaluation sample as memorisable if it is similar in terms of input and output token overlap to one or more training samples e.g. an evaluation sample consisting of the input "What is a tool for indicating air pressure? (A) seismograph (B) barometer

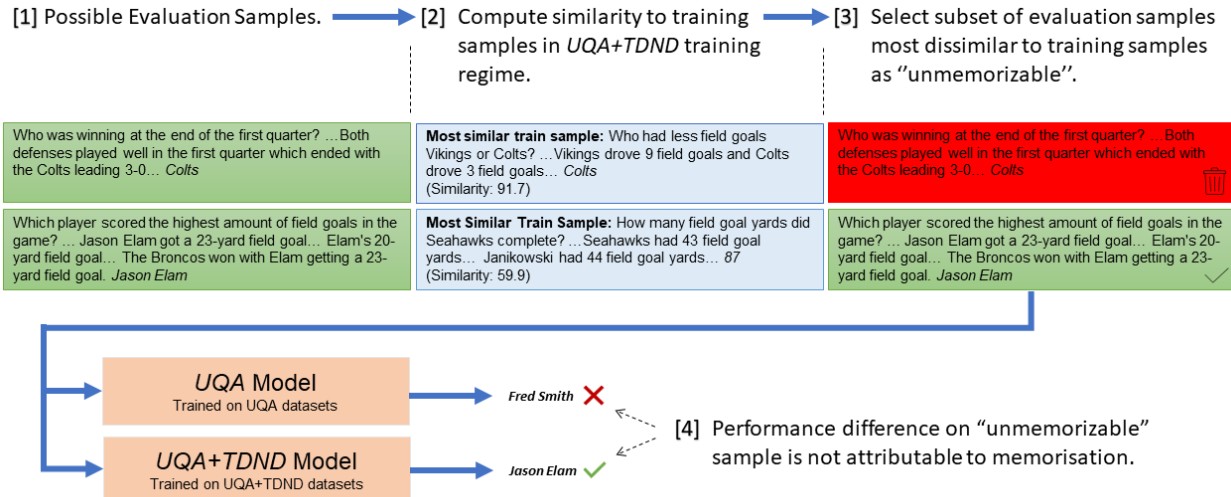

Figure 1: Visualisation of key aspects of our methods. We consider two models, one trained on a set of question-answering datasets (*UQA*) and the other trained on *UQA* plus two additional datasets (*UQA+TDND*). *TDND* samples are constructed so as to improve performance on some of our evaluation datasets and to be irrelevant for others. Our objective is to understand whether any improvement is attributable to memorisation or to *TDND* samples imparting an improved ability to generalise. We select evaluation samples that are very unlikely to have become memorisable from our training datasets based on a semantic similarity score (section 2.3), and compare performance between the two models. Our method enables evaluating performance for each model on the *same* subset of unmemorisable samples, and it does not require access to the pretraining corpus.

..." and label "barometer" is memorisable if a sample with input "Which weather instrument measures air pressure? (A) barometer (B) rain gauge ..." and label "barometer" exists in the training data. To identify memorisable evaluation samples we propose a method of scoring similarity between each evaluation and each training sample using semantic similarity as encoded in sentence embedding vectors produced by a Sentence Transformers model (Reimers & Gurevych, 2019). This is discussed in more detail in section 2.3.

The UnifiedQA project (*UQA*) (Khashabi et al., 2020) demonstrated that it is possible to attain good performance on unseen evaluation datasets (those that have not been involved in training) after further training of a pretrained Language Model on a variety of question-answering datasets in a multitask fashion. One of the unseen RC datasets that Khashabi et al. (2020) use for evaluation is DROP (Dua et al., 2019). Performance on DROP is rather poor in the *UQA* setting. This dataset requires simple numerical literacy in order to correctly answer a question. A separate study, Geva et al. (2020), demonstrated significant performance improvement on DROP by pretraining on two synthetic datasets (collectively referred to here as *TDND*) that they designed to impart simple numerical reasoning strategies. We add *TDND* to the *UQA* training mixture (denoted *UQA+TDND*) and analyse the impact on subsets of DROP (Dua et al., 2019), ROPES (Lin et al., 2019), and several other unseen RC and MC datasets that are unlikely to be memorisable, even after the addition of the *TDND* datasets.

In summary the major contributions of this paper are: (A) We propose a method of identifying evaluation-train overlap based on semantic similarity of input and output tokens. (B) We propose a method to intervene with additional training datasets versus a baseline, both to mitigate effects of pretraining on results and to avoid the need to compare disparate populations of evaluation subsets. (C) We demonstrate the effectiveness of our methods in identifying both memorisable, and unmemorisable samples. (C) We show that performance on unmemorisable subsets of DROP and ROPES is significantly improved by the addition of *TDND* training datasets.

## 1.1 Related Work

As in our case, prior work on studying the effects of memorisation on model performance in the NLP domain has generally focused on identifying subsets of evaluation data that are either unlikely or likely to have been memorised from training data and considering the performance of the subset in conjunction with the nature of the input samples. Lewis et al. (2021) consider open-domain single-hop factual questions. By identifying test questions with answers matching training questions and then manually identifying those evaluation samples where the question is or isn't a paraphrase of a training question, they show that smaller Language Models (such as the BART model (Lewis et al., 2020) we also use) exhibit low performance on samples that don't have a match in the training set. Our paper can be considered as an extension of this work in the area of RC questions that require reasoning over a context to answer. We show that in contrast to their findings on factual questions, a BART model is capable of *improved* performance for RC samples without a memorisable match in the training set. Elangovan et al. (2021) consider train-test overlap on different NLP tasks to ours. To evaluate similarity they utilise cosine similarity between sparse bag-of-words vectors constructed for each test and train sample. Similar to our study, a recent work, Kambhatla et al. (2023), considers cosine similarity over sentence embedding vectors as the similarity measure, although they differ from our purpose in that they focus on identifying dataset contamination between test and train splits within the same dataset, and in other methodological aspects such as controlling for the effects of pretraining as discussed further in section 2.

The effect of evaluation dataset contamination in the pretraining datasets of large Language Models (LLMs) has been reported in a number of studies (Brown et al., 2020; Sanh et al., 2021; Wei et al., 2021; Du et al., 2022; Chowdhery et al., 2022). These generally automate the process of contamination discovery by considering $n$-gram overlap between evaluation datasets and pretraining data. A filtered, clean version of each evaluation dataset is sometimes then constructed and performance is compared to that of the full dataset. Generally these studies find that even where an evaluation dataset is found to heavily overlap with pretraining data, the performance gap between clean and full versions is small and each clean version may either slightly underperform or slightly overperform the full version. Although we are not disagreeing with the overall findings, one criticism of this approach is that $n$-gram overlap can only detect test-train overlap where the overlap is an exact match of contiguous tokens, while paraphrases or overlaps between discontinuous tokens that otherwise overlap highly will not be detected.

Also focusing on memorisability in pretraining data in the situation where the pretraining corpus is available, Carlini et al. (2023) evaluate memorisation by prompting a model with a particular sequence and ascribing memorisation if the model continuation is an exact match to the ground truth continuation of that sequence. They show that the degree of memorisation increases both with the size of the model and with the number of duplicates of a sequence in the pretraining data. Lee et al. (2022) show that training on de-duplicated pretraining data results in memorised text being generated ten times less frequently. Kandpal et al. (2023) show that single-hop factual question answering performance is correlated with the number of documents containing the question and answer entities seen in pretraining. In the domain of numerical reasoning, Razeghi et al. (2022) show that numerical term frequency in the pretraining corpus also correlates with accuracy. The study goes on to remove evaluation samples that are likely to have been memorized i.e. those where the input terms and the answer co-occur in a pretraining document. It was then found that the performance of the remaining unmemorisable samples continues to correlate with the frequency of the input terms in the pretraining corpus, suggesting that the performance improvement is not solely due to memorisation.

As a reminder that *spurious* memorisation can lead to lower results in downstream evaluation as well as inflating results, Hartill et al. (2023) show that removing near-duplicate Musique (Trivedi et al., 2022) training samples from a BART model training regime resulted in improved downstream performance where evaluation samples had input token overlap with the duplicated training samples but had different labels.

Outside of the NLP domain, a number of studies have challenged the historical assumption that an ability to memorise the training set and an ability to generalise are mutually exclusive (Zhang et al., 2021). In considering overparameterised models (those with more trainable parameters than samples they are trained on), Zhang et al. (2017) found that such models are capable of perfectly memorising a training set with

randomly assigned labels, without learning any ability to generalise. Models trained on the same training data except with correct labels assigned are of course able to generalise successfully to test samples. By varying the degree of randomness in assigning labels to training samples between these two extremes the authors found a correlation between generalisation error and the amount of label noise, showing that overparameterised neural networks are capable of both capturing the extant signal in the data, while at the same time memorising the noisy part. Feldman (2019) proposes that memorisation in long-tail distributions (i.e. the common case where classes consisting of small numbers of samples collectively comprise a significant fraction of the distribution) is actually necessary in minimising generalisation error, and empirically demonstrates this in Feldman & Zhang (2020). The focus of our study differs from these in that we are primarily interested in evaluating whether a model in our setting can exhibit an ability to generalise in the absence of an opportunity to memorise.

With more distant connection with our work, Hacohen et al. (2020) show that various neural models learn similar classification functions at particular stages of training. Exploring this idea in the NLP domain, Choshen et al. (2022) study the order that linguistic phenomena are learned over the course of training and find that neural Language Models with differing architecture and training data tend to acquire particular linguistic abilities in a similar order. Future work might consider the relationship, if any, between such order of learning and the acquisition of skills involving memorisation versus those relating to more abstract RC skills such as logical operations, multi-step reasoning and so forth.

## 2    Method

In the context of language models, Carlini et al. (2023) characterise memorisation as the generation of an exact continuation of a text sequence, given the first part of the sequence as input. Several other studies (section 1.1) test for potential memorisation (evaluation dataset contamination) as the presence of $n$-gram(s) in training samples that co-occur in evaluation samples (where $n \geq 8$). In this paper we take a view of potential memorisation as occurring where there is not only overlap in a *contiguous* sequence of tokens but also where a *discontinuous subset* of input tokens could directly produce a particular output. For example learning one or more training samples similar to "Who had more field goals Vikings or Colts? ..." with label "Colts" could cause a model with evaluation input "Who was winning at the end of the first quarter? ... Colts leading 3-0..." to predict "Colts" without any semantic understanding of the question or the context. We develop an alternative method of evaluating evaluation-train similarity using cosine similarity of evaluation and train sample sentence embedding vectors. We find that this approach surfaces test-train overlaps where the tokens discontinuously (or contiguously) overlap (see section 2.3).

In some prior work it has been necessary to compare disparate populations of evaluation samples in order to draw conclusions. For example Chowdhery et al. (2022) note that in comparing the full version of an evaluation dataset to a filtered version consisting only of unmemorisable samples they are comparing different subsets. We address this issue by identifying evaluation samples that will not be rendered memorisable by the addition ("intervention") of new training datasets and then using this same subset to evaluate the performance difference before and after our intervention. This approach has the added benefit that we do not need access to the pretraining corpus. A visual overview of our approach is provided in Figure 1.

Below we discuss how the training regimes for our "before" model ($UQA$) and "after" model ($UQA+TDND$) are constructed, our evaluation datasets, and our methods for identifying evaluation samples that are very unlikely to have become memorisable by the intervention of the additional training datasets.

### 2.1   *UQA* and *UQA+TDND* Model Training

Our main experiments evaluate the performance difference between two models; $UQA$ and $UQA+TDND$. Both are trained using the same hyperparameters (appendix A), the only differences being the respective sets of datasets used to train them. We experimented with differing combinations of hyperparameters on both training mixtures until we found a set that worked well over both. Training is performed in a multi-task manner, uniformly sampling over the training datasets. The best model from each run is selected as that with the highest mean performance over all development sets after 150,000 train steps which allows for some

Table 1: Best model selection for three runs each of *UQA* and *UQA+TDND*. Step is the training step at which the best model is selected. Dev Perf is the mean accuracy over constituent development sets. The *UQA+TDND* best model has usually but not always been trained for more steps than the *UQA* best model.

| | **UQA** | | **UQA+TDND** | |
|---|---|---|---|---|
| **Run** | **Step** | **Dev Perf.** | **Step** | **Dev Perf.** |
| 1 | 140,000 | 65.80% | 150,000 | 67.45% |
| 2 | 110,000 | 66.62% | 140,000 | 68.76% |
| 3 | 140,000 | 66.13% | 140,000 | 68.74% |

flexibility in tuning per training mixture as shown in Table 1. We make use of a similar fixed prompting format to Khashabi et al., 2020; 2022 (appendix B), and take as our *UQA* baseline the same set of training datasets that they use. Specifically, *UQA* consists of datasets of RC type; SQUAD 1.1 (Rajpurkar et al., 2016), SQUAD 2 (Rajpurkar et al., 2018), NarrativeQA (Kočiský et al., 2018), along with MC datasets RACE (Lai et al., 2017), ARC (Clark et al., 2018), Regents (Clark et al., 2016) ("Sci-Elem" and "Sci-Mid" in this paper) , OpenbookQA (Mihaylov et al., 2018), MCTest (Richardson et al., 2013), and one binary-labelled dataset, BoolQ (Clark et al., 2019).

As noted, Geva et al. (2020) developed two synthetic datasets designed to impart numerical reasoning ability of the sort needed to improve model performance on DROP (Dua et al., 2019). Of these, "Textual Data" (*TD*) contains RC samples with similar vocabulary and involving similar reasoning skills to DROP (e.g. "Who had the lowest number of field goal yards in total? ... Dolphins nailed 26 field goal yards and Vikings nailed 15 field goal yards...", label "Vikings"). The second dataset, "Numerical Data" (*ND*) contains a large number of samples with inputs consisting of symbolic expressions (e.g "argmin(undergrass 11952 bussu 3315)?", label "bussu"). Geva et al. (2020) show that pretraining on *TD* and *ND* followed by finetuning on DROP leads to substantially higher performance. In our case, we convert the datasets (collectively *TDND*) to our format and add them to the *UQA* training mixture to train our *UQA+TDND* model.

## 2.2 Unseen Evaluation Datasets

We selected evaluation datasets as follows:

**DROP** (Dua et al., 2019) is a RC dataset designed to test numerical reasoning. Questions are created by human annotators on paragraphs sourced from Wikipedia.

**DROP-CS** (Gardner et al., 2020) contains perturbed versions of DROP Test split samples e.g. by making a minor change to the context such that the label is changed.

**ROPES** (Lin et al., 2019) is a RC dataset that requires multi-step reasoning over a situation, often involving qualitative relations such as "higher" or "lower". Questions are human-authored based on passages from Wikipedia and science textbooks.

**NewsQA** (Trischler et al., 2017) is a RC dataset of human-authored questions about CNN articles.

**PIQA** (Bisk et al., 2020) is a two-option MC dataset covering physical commonsense questions. Samples are created by human annotators from prompts sourced from `instructibles.com`.

**CSQA** i.e. CommonsenseQA (Talmor et al., 2019) is a five-option MC dataset of commonsense questions derived from Conceptnet (Speer et al., 2017).

**QASC** (Khot et al., 2020) is an eight-option MC dataset covering human-authored science questions that require two facts to answer. Facts are sourced from a corpus derived from open web pages (Clark et al., 2016).

In all cases we started with the publicly available development split. We discovered that the DROP development set contained over 800 exact duplicates. Because we were unsure whether duplicate samples were the result of some bias in dataset creation that could manifest itself when we select smaller "unmemorisable"

subsets we de-duplicated all our evaluation datasets and note that DROP-CS also contained a very small number of duplicates. An example for each dataset is shown in table 4.

When selecting "unmemorisable" subsets (see section 2.3 below) we observed that samples with numeric answers were much more likely to be filtered out since many such answers tend to be commonly occurring small numbers (1, 2, 5...). To combat this bias we remove all samples with numeric answers from our DROP and DROP-CS evaluation.

The resulting sample counts are in Table 2. Elaboration as to how the "Least similar" and "Unmemorisable" subsets are derived follows in the next section.

Table 2: Evaluation Dataset sample counts. "All" is the total sample count after de-duplication and removal of samples with numeric answers. Least Similar is the subset of these with a Similarity Score of each evaluation sample to it's most similar training sample under 60.0. Unmemorisable samples are those Least Similar which also have no answer term overlap with the most similar training sample.

| Eval Dataset | All | Least Similar | Unmemorisable |
|---|---|---|---|
| DROP | 3277 | 867 | 652 |
| DROP-CS | 478 | 154 | 110 |
| ROPES | 1688 | 307 | 197 |
| NewsQA | 4341 | 1204 | 759 |
| PIQA | 1838 | 1354 | 588 |
| CSQA | 1221 | 233 | 129 |
| QASC | 926 | 139 | 99 |

## 2.3 Similarity Computation Method

To evaluate similarity between evaluation and training samples, we use sentence embedding vectors produced by the "sentence-transformers/stsb-roberta-large" model (Reimers & Gurevych, 2019) from the Huggingface library (Wolf et al., 2020). We quantify the "memorisability" of each evaluation sample from each training sample by computing a Similarity Score as:

$$sim(e_i, t_j) = \frac{csim(e_i^q, t_j^q) + csim(e_i^a, t_j^a)}{2} * 100 \tag{1}$$

Here $e_i$ and $t_j$ are the embeddings for the $ith$ evaluation and $jth$ training samples, $q$ and $a$ refer to the question (including context) and answer components of each, and csim is the cosine similarity function. We consider both $q$ and $a$ equally as we are primarily interested in identifying evaluation-train pairs where a memorised answer could inflate results. Alternative formulations that consider $q$ only would also identify spuriously memorisable samples that could deflate results but this does not suit our purpose here.

We require a memorisability threshold $T$ for Similarity Scores, below which sample pairs are sufficiently dissimilar as to be unmemorisable. The choice of a value for $T$ involves a trade-off between confidence that no memorisable samples remain and diminishing sample counts. We identified a suitable value of $T$ through an iterative process of humanly evaluating the ten most similar sample pairs for each evaluation dataset at a possible value for $T$ and increasing this value at each iteration until we found a value at which no memorisable sample pairs were identified but remaining sample counts are reasonable (Table 2). This value was identified as $T = 60$. We cross-checked this by searching for the lowest Similarity Score for any sample pair where we considered the evaluation sample to be memorisable. This value was found to be substantially higher than 60, further increasing our confidence that evaluation subsets identifed at $T = 60$ were unlikely to contain memorisable samples (the most similar pair for each subset at $T = 60$ is shown in appendix D). We call the resulting subset of samples for each evaluation dataset "Least Similar".

Table 3: In-domain Test-Train Overlap. Most similar test-train pairs for each constituent training dataset as measured by Similarity Score (in brackets). The actual evaluation split used is in square brackets. For readability, multi-choice options are removed, remaining context is truncated and answers are in *italics*. The same pair was identified in both SQuAD 1.1 and SQuAD 2 hence shown once. Train samples that are identical or paraphrases to evaluation samples from the same dataset are highlighted in red.

| Dataset | Eval Sample [Split] | Most Similar Train Sample |
|---------|---------------------|---------------------------|
| Sci-Elem | Green plants get the energy they need to make food from? *sunlight* [Test] | Identical except for order of multi-choice options. (99.48) |
| Sci-Mid | Iron oxides such as rust form when iron metal reacts with oxygen in the air. What are the chemical symbols for the two elements found in iron oxide? *Fe and O* [Test] | Identical. (100.00) |
| ARC-Easy | Which of the following elements is best able to combine with itself and hydrogen [H] to form large molecules? *carbon [C]* [Test] | Identical. (100.00) |
| ARC-Hard | Students watched a bird fly to and from a large bush every few minutes. The students told their teacher "The bird has a nest in that bush." This statement is an example of? *an inference made from observations* [Test] | Identical except that one multi-choice option is different. (99.91) |
| BoolQ | Has an mlb game ever ended in a tie? ... The longest game by innings in Major League Baseball was a 1–1 tie... *Yes* [Dev] | Identical. (100.00) |
| MCTest | What did Hannah and Mary chase at the park? ... Hannah and Mary ran around chasing butterflies for a little time... *butterflies* [Dev] | What did my granddaughter try to catch? ... granddaughter Tina ... catch ... butterfly... *butterfly* (87.53) |
| OBQA | Oak tree seeds are planted and a sidewalk is paved right next to that spot until eventually the tree is tall and the roots must extend past the sidewalk which means? *parts may break the concrete* [Test] | Identical except for order of multi-choice options. (99.95) |
| RACE | The last sentence in the passage shows that _ ? ... Little Tommy ... said "Well on the first day of school when I saw that man nailed to the plus sign I knew they weren't joking. " *Tommy was afraid of being nailed* [Test] | Identical. (99.99) |
| SQuAD | Under Elie Metchnikoff's cellular theory what cells were responsible for immune response? ... According to the cellular theory of immunity ... by Elie Metchnikoff it was ... phagocytes... *phagocytes* [Dev] | Question is a paraphrase ("Cellular immunology expressed the theory that what cells caused immune responses?"), context and answer are identical. (99.75) |

Acknowledging the possibility that some number of Least Similar samples could still be memorisable we then took a further subset of Least Similar samples where the answer has no word overlap with the most similar training sample. For brevity we call this further subset "Unmemorisable" as shorthand for "unlikely to be memorisable from our training datasets, including *TDND*". We note that we are unable to eliminate evaluation samples that have answer overlap with *any* training sample as this would eliminate too many samples. It is also worth clarifying that our definition of "Unmemorisable" does not preclude a given evaluation sample being memorisable from pretraining data. Since we are comparing performance before and after the intervention with *TDND* datasets it is only strictly necessary that our Unmemorisable samples not be memorisable from *TDND* although in practice we ensure they are not memorisable from any of our *UQA+TDND* datasets.

### 2.3.1 Similarity Computation Evaluation - In-Domain Datasets

We initially evaluate the calibration of our method by considering similarity between the train and development/test splits of our training datasets. As Table 3 shows, identical or near identical sample pairs occur for most training datasets and these tend to score close to 100.

### 2.3.2 Similarity Computation Evaluation - Evaluation Datasets

Turning to our evaluation datasets, we first consider the most similar overall eval-train pair for each evaluation dataset. Generally we find the incidence of identical or near identical pairs is much lower than is the case for the above in-domain evaluation, however memorisable evaluation samples certainly exist as shown in

Table 4: Overlap between unseen evaluation and train datasets. Most similar overall sample pair for each evaluation dataset as measured by Similarity Score (in brackets). For readability, multi-choice options are removed, remaining context is truncated and answers are in italics. Red denotes train samples that could potentially make the corresponding evaluation sample memorisable through contiguous or discontiguous sets of input tokens.

| Eval Dataset | Eval Sample | Most Similar Train Sample |
|---|---|---|
| DROP | Which household was second most common? ... there were 19306 households ... 39.9% were non-families... *non-families* | SQuAD 1.1: What is the second highest demographic for households? ... There were 230233 households ... 37.4% were non-families... *non-families* (94.40) |
| DROP-CS | Which team went scoreless in the third quarter? ... Buffalo ... connected ... 8-yard TD pass for the only score of the period... *Vikings* | TD: Who had the lowest number of field goal yards in total? ... Dolphins nailed 26 field goal yards and Vikings nailed 15 field goal yards... *Vikings* (89.96) |
| ROPES | Will Seattle have more or less sulfur oxides in the air than St. Louis? ... Seattle has installed a new wind farm and zero emission solar farm to generate power while St. Louis recently installed a coal fired power plant ... *less* | SQuAD 1.1: Were sulfonamides more or less toxic than arsphenamine? ... Compared to arsphenamine the sulfonamides ... were far less toxic ... *less* (81.13) |
| NewsQA | What was the score in the Werder Bremen Athletic Bilbao game? ... Werder Bremen beat Athletic Bilbao 3-0 ... *3-0* | SQuAD 2: What was the winning score for the game with Real Madrid at Bernabeu stadium? ... The pinnacle of the ... season ... the ... Bernabéu Stadium in a 3–0 win over Real Madrid... *3-0* (88.06) |
| PIQA | Trees? *provide homes for animals* | RACE: The story is about _ ? ... Some animals live in holes in trees ... *the homes of some animals* (77.04) |
| CSQA | The water in clouds turn in to what when it gets cold? *snowflake* | ARC-Hard: Which form of water is most likely to appear when the temperature is below freezing? *snow* (87.27) |
| QASC | What is a tool for indicating air pressure? *barometer* | Sci-Elem: Which weather instrument measures air pressure? barometer (95.14) |

Table 4. In contrast to the above in-domain evaluation where contiguous overlaps of tokens in similar pairs are common, it can be seen that memorisable samples in Table 4 generally would not have been detected without a method that can pick up discontinuous token overlaps.

For brevity, the supporting table of Least Similar evaluation-train pairs is in appendix D, having already noted that we cannot identify any memorisable evaluation samples in that category. Similarly, Appendix E shows the most similar evaluation-train pair for Unmemorisable evaluation samples. Unsurprisingly we cannot identify any memorisable evaluation samples here either.

## 3 Main Experiment

All evaluation datasets of RC format are evaluated using the F1 score as formulated by Rajpurkar et al. (2016). The MC datasets are evaluated by taking the option with the highest overlap with the predicted answer and then scoring as exact match.

The *UQA* and *UQA+TDND* Models are based on BART (Lewis et al., 2020). All models use the Huggingface (Wolf et al., 2020) implementations. We train three models for each of *UQA* and *UQA+TDND* respectively using different random seeds and take the mean over each set as our main measure. We ascribe statistical significance to performance change between *UQA* and *UQA+TDND* if it is at the 95% confidence level (confidence intervals and standard deviations are in appendix C).

### 3.1 Experimental Results and Discussion

Table 5 shows the effect of adding the *TDND* datasets to the training regime. Considering the unfiltered evaluation sets comprised of "All Samples", it is no surprise that DROP and DROP-CS show a large per-

Table 5: Effect of intervention with *TDND* datasets on All, Least Similar, and Unmemorisable evaluation samples. Figures are the mean over three model runs trained with different random seeds. Statistically significant changes at the 95% confidence level are marked in **bold** i.e. the improvement for DROP and ROPES is significant in Least similar and Unmemorisable subsets, changes for other datasets are not.

| Eval Dataset | Random | All Samples | | | Least Similar | | | Unmemorisable | | |
| | | UQA | UQA +TDND | % Change | UQA | UQA +TDND | % Change | UQA | UQA +TDND | % Change |
|---|---|---|---|---|---|---|---|---|---|---|
| DROP | | 40.2 | 46.5 | **15.7** | 41.0 | 43.9 | **7.1** | 41.7 | 45.5 | **9.0** |
| DROP-CS | | 32.0 | 38.2 | **19.3** | 36.3 | 41.8 | 15.3 | 38.5 | 42.2 | 9.6 |
| ROPES | | 41.2 | 51.9 | **26.1** | 46.5 | 55.3 | **18.9** | 41.9 | 52.6 | **25.7** |
| NewsQA | | 57.3 | 56.6 | -1.3 | 52.8 | 50.3 | -4.7 | 53.4 | 51.4 | -3.7 |
| PIQA | 50.0 | 63.5 | 62.3 | -1.9 | 62.2 | 61.7 | -0.8 | 60.3 | 60.4 | 0.1 |
| CSQA | 20.0 | 55.6 | 55.4 | -0.4 | 61.5 | 61.2 | -0.5 | 60.7 | 61.0 | 0.4 |
| QASC | 12.5 | 37.7 | 36.2 | -3.8 | 35.7 | 34.1 | -4.7 | 36.4 | 33.7 | -7.4 |

formance improvement (15.7% and 19.3% respectively) since the *TDND* datasets are specifically designed for that purpose. Moving to the Unmemorisable subsets, there is still a 9% performance improvement for DROP showing that while there is some diminishment, a material performance improvement that is not attributable to memorization remains. DROP-CS improvement is similar but this result is not significant due to the small sample size. While our experiment cannot tell us what mechanism is responsible for this ability to generalise, the intuitive explanation is that *TDND* datasets have as intended imparted relevant numerical reasoning strategies.

ROPES shows an even larger performance improvement than DROP over All Samples which is largely retained for the unmemorisable subset (26.1% → 25.7%). Noting that like DROP, ROPES also requires multi-step reasoning over a context and often involves qualitative relations like "less" or "lower" (Lin et al., 2019) it is reasonable to say that benefits imparted by *TDND* samples are responsible for the improvement. For example a typical *TD* sample might involve a judgement such as "Who had the lowest number of field goal yards in total? ... Dolphins nailed 26 field goal yards and Vikings nailed 15 field goal yards..."

### 3.2 Limitations

Since our similarity computation (equation 1) considers both the question and the answer components it is able to identify evaluation samples that contribute to *inflated* results from the model emitting memorised but correct answers. However using the equation 1 formulation, we cannot say what could be *deflating* results (e.g. NewsQA and QASC in Table 5). For example, it could be an effect of spurious memorisation where an incorrect answer is emitted based on one or more superficially similar training samples, random perturbation, or it could equally be some other factor such as the result of the incorrect application of some method learned as a result of the *TDND* intervention.

## 4 Conclusion

We have proposed a method of identifying evaluation-train overlap based on semantic similarity of input and output sequences that is reinforced by the further elimination of evaluation samples with overlap in answer terms to the most similar training sample. We have shown that this method is able to identify evaluation samples that are memorisable through both contiguous and non-contiguous token overlap with similar training examples.

To avoid the pitfall of having to compare disparate populations of evaluation samples, as well as to eliminate any dependency on knowing the contents of the pretraining dataset, we have also proposed a method for determining whether or not performance improvement is attributable to memorisation. This involves an intervention through the addition of training datasets that might be expected to improve performance on some evaluation datasets but not on others and measurement of the resulting performance difference. We

have shown that for contextualised questions there is significant performance improvement on unmemorisable subsets of DROP and ROPES i.e the improvement is not attributable to memorisation.

**Broader Impact Statement**

The effect of memorisation can be both positive and negative. For example it may be desirable for a model to memorise answers to very specific factual questions encountered in training, while as in our case, it is undesirable for it to emit memorised answers to questions where the intent is for a model to "reason" to an answer based upon supplied contextual information. For this reason while we believe that our methods have a number of applications, due consideration should be given to the interpretation of what they uncover.

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

## A   Hyperparameters

All models are trained on two Nvidia RTX8000 GPUs using 32-bit precision and a linear learning rate decay schedule that reduces the learning rate to zero over 250K training steps. Initial learning rates and other hyperparameters are shown in Table 6. The optimiser used is AdamW. A maximum sequence length of 512 tokens was used for all models.

Table 6: Hyperparameters used for each model. Each training step is one batch input i.e the number of optimization steps is $TrainingSteps/GradientAccumulationSteps$. All final models are selected as the best model on the development sets over the specified number of training steps and validation steps were performed every 10K training steps.

| Model | Initial LR | Batch Size | Grad. Accum | Train Steps |
|---|---|---|---|---|
| *UQA* Models | 2e-5 | 32 | 2 | 150K |
| *UQA+TDND* Models | 2e-5 | 32 | 2 | 150K |

## B   QA Model Input Format

Our input format is similar to that used in UnifiedQA (Khashabi et al., 2020). The only modification is to add an open domain format to accommodate the *ND* dataset.

Open domain form:
`[question] \\n`

Reading comprehension (RC) form:
`[question] \\n [context]`

Multiple choice form:
`[question] \\n (A) [option text a] (B) [option text b] ...`

Multiple choice with RC form:
`[question] \\n (A) [option text a] (B) [option text b] ...  \\n [context]`

## C   Means, Standard Deviations and 95% Confidence Intervals

Table 7: Mean (Standard Deviation) and 95% Confidence Interval for each set of model runs. Confidence Intervals (CI) are constructed for the difference of the corresponding *UQA* and *UQA+TDND* means.

| Eval Dataset | All Samples | | | Least Similar | | | Unmemorisable | | |
|---|---|---|---|---|---|---|---|---|---|
| | UQA | UQA +TDND | 95% CI | UQA | UQA +TDND | 95% CI | UQA | UQA +TDND | 95% CI |
| DROP | 40.2 (1.0) | 46.5 (1.0) | (-7.676, -4.922) | 41.0 (1.8) | 43.9 (2.0) | (-5.647, -0.177) | 41.7 (1.3) | 45.5 (2.2) | (-6.960, -0.544) |
| DROP-CS | 32.0 (3.7) | 38.2 (2.5) | (-9.062, -3.306) | 36.3 (4.2) | 41.8 (3.4) | (-11.306, 0.208) | 38.5 (4.2) | 42.2 (3.9) | (-10.911, 3.553) |
| ROPES | 41.2 (1.7) | 51.9 (3.1) | (-12.692, -8.754) | 46.5 (3.5) | 55.3 (6.5) | (-14.048, -3.545) | 41.9 (1.7) | 52.6 (6.2) | (-16.659, -4.838) |
| NewsQA | 57.3 (1.3) | 56.6 (0.9) | (-0.933, 2.480) | 52.8 (2.4) | 50.3 (1.9) | (-0.489, 5.475) | 53.4 (2.1) | 51.4 (1.6) | (-1.804, 5.791) |
| PIQA | 63.5 (0.8) | 62.3 (0.5) | (-1.670, 4.136) | 62.2 (1.1) | 61.7 (0.9) | (-2.845, 3.780) | 60.3 (1.9) | 60.4 (1.2) | (-4.933, 4.820) |
| CSQA | 55.6 (1.3) | 55.4 (0.1) | (-2.902, 3.339) | 61.5 (0.4) | 61.2 (2.5) | (-7.619, 8.192) | 60.7 (0.4) | 61.0 (4.1) | (-10.761, 10.244) |
| QASC | 37.7 (1.0) | 36.2 (0.7) | (-0.988, 3.868) | 35.7 (2.9) | 34.1 (0.9) | (-4.263, 7.621) | 36.4 (3.8) | 33.7 (2.7) | (-4.489, 9.876) |

# D Most Similar Evaluation-Train Pairs within Least Similar Evaluation Samples

Table 8 shows the most similar evaluation-train pair for each of our Least Similar evaluation subsets.

Table 8: Overlap between Least Similar evaluation dataset subsets and train datasets. Most similar sample pair for each Least Similar subset as measured by similarity score (in brackets). For readability, multi-choice options are removed, remaining context is truncated and answers are in italics.

| Eval Dataset | Eval Sample | Most Similar Train Sample |
|---|---|---|
| DROP | Which racial group made up the least of the country? ... The racial makeup of the county was 81.2% white 12.7% black or African American 2.4% Asian 0.3% American Indian 0.1% Pacific islander ... *Pacific islander* | SQuAD1.1: Where was the coconut palm brought to St. Barts from? ... Coconut palm was brought to the island from the Pacific islands... *the Pacific islands* (59.99) |
| DROP-CS | Which player caught the shortest TD pass? ... Tomlinson getting a 3-yard TD pass to Philip Rivers... *Philip Rivers* | TD: How many field goal yards did Dolphins Jaguars' quarterback and Bears have combined? ... 13 field goal yards ... 53 field goal yards ... 57 field goal yards *123* (59.99) |
| ROPES | What hour did storage costs go up: 1 PM or 3 PM? ... the access times go up as more data is read CPU load goes up as XML data takes more power to process and storage costs go up. ... At 1 PM he stored 1 Gigabyte ... At 3 PM he didn't store anything... *1 PM* | TD: How many more passes did Houston have than impressive wins ? ... Houston drove 6 passes... Houston drove 5 impressive wins... *1* (59.97) |
| NewsQA | Which series inspired the popularity of the name Cullen? ...The boy's name that rocketed up the list the fastest is Cullen – the name of the lead character in the popular "Twilight" book series... *"Twilight"* | SQuAD1.1: At the time of release which episode of the Legend of Zelda series was considered the greatest entry? ... Twilight Princess was considered the greatest entry in the Zelda series... *Twilight Princess* (59.98) |
| PIQA | Make homemade pasta from dough? *Roll out the dough so that is thin and take a knife and cut slices from the dough to make individual pieces and put it in a pot to boil.* | Sci-Mid: In making a pizza which process involves a chemical change? *baking the dough to form the crust* (59.99) |
| CSQA | She wanted a kitten and puppy so why did she only get the puppy? ... *one choice for pet* | RACE: The article is most likely intended for _ ? Animal shelters are full of dogs cats rabbits and more animals all in need of loving homes... *pet lovers* (59.95) |
| QASC | What must be done to classify minerals? *scratch them* | ND: What is argmin(duco 14490.16 silvanus 16272 scratchification 3156.6)? *scratchification* (59.92) |

# E  Most Similar Evaluation-Train Pairs within Unmemorisable Evaluation Samples

Table 9 shows the most similar evaluation-train pair for each of our Unmemorisable evaluation subsets.

Table 9: Overlap between Unmemorisable evaluation dataset subsets and train datasets. Most similar sample pair for each Unmemorisable subset as measured by similarity score (in brackets). For readability, multi-choice options are removed, remaining context is truncated and answers are in italics.

| Eval Dataset | Eval Sample | Most Similar Train Sample |
|---|---|---|
| DROP | Of the languages listed which are spoken by fewer than 3000 people? ... Other languages include ... Tagalog language with 2888 ... Japanese with 2546 and African languages with 2546 *Tagalog Japanese African languages* | SQuAD 1.1: What is Oklahoma's fourth most popular language? ... German is the fourth most commonly used language with 13444 speakers *German* (59.98) |
| DROP-CS | Which player caught the shortest TD pass? ... Tomlinson getting a 3-yard TD pass to Philip Rivers... *Philip Rivers* | TD: How many field goal yards did Dolphins Jaguars' quarterback and Bears have combined? ... 13 field goal yards ... 53 field goal yards ... 57 field goal yards *123* (59.99) |
| ROPES | What time did storage costs go up: 7 PM or 6 PM? ... At 6 PM he got dinner. At 7 PM he stored 55444 Gigabytes ... *7 PM* | RACE: From the text we can infer this article was probably written in __ ? ... The award is given every two years. The next one will be given in 2008 *2007* (59.96) |
| NewsQA | Who is missing? ... Authorities are searching for a female soldier missing after a fire at her apartment ... 2nd Lt. Holley Wimunc ... *Lt. Holley Wimunc* | NarrativeQA: Who was the second man that was out on the moors the same time as Sir Henry and Watson? ... Watson tracks the second man he saw in the area and discovers it to be Holmes ... *Sherlock Holmes* (59.97) |
| PIQA | How do you power through something? *keep going no matter what* | ND: What is argmax(foremostly 11886.1 continuousness 16062.42 matchable 5062.8 washout 1295)? *continuousness* (59.99) |
| CSQA | The end of the barrel of what primitive firearm is bell shaped? *blunderbuss* | ND: What is argmin(undergrass 11952 bussu 3315)? *Bussu* (59.95) |
| QASC | What must be done to classify minerals? *scratch them* | ND: What is argmin(duco 14490.16 silvanus 16272 scratchification 3156.6)? *scratchification* (59.92) |

