# OpenReview forum: "Do Smaller Language Models Answer Contextualised Questions Through Memorisation Or Generalisation?"
_TMLR — Rejected by TMLR_

### Review · Reviewer_AAt4 · 2023-10-16

**Summary Of Contributions:**

The paper proposes to compare train and test examples with SBERT (+ word overlap heuristic) to assess whether an evaluation example might be memorised from the train set.
The paper replicated previous results that adding the TDND pair of datasets improves results, even when removing overlaps. (The results drop, but are still higher than the baseline).

**Audience:**

Yes

**Claims And Evidence:**

Yes

**Requested Changes:**

# writing
See weaknesses
# references:
Papers discussing the order of learning (e.g. The grammar-learning trajectories of neural language models‏, and Many of Guy Hacohen's non NLP works) often show that models learn in the same order, which might have a different prediction than more memorisation in larger ones, "just so" but memorisation when stopped not memorizing. This relates to noisy data learning, and to the long-tail hypothesis, which states memorization is itself learning See Feldman's works (e.g. What Neural Networks Memorize and Why: Discovering the Long Tail via Influence Estimation)
# Minor:
When you cite someone in line, use \citet{} (for example in the intro at e.g. (Carlini...) Should be e.g., Carlini (20...) or "For example (Chowdhery et al., 2022)" )
Also consider using \citet[before;][after] or \citealp in cases of parenthesis just before\after the citations (e.g., Khashabi et al. (2020; 2022) (appendix B),).
Fig 1 is not easily interpretable + it has too many colors and styles, which instead of making the most important things pop, it makes everything compete on attention and just feel cluttered.
Why 100 **.0** and 2 **.0**?

**Strengths And Weaknesses:**

# Strength
The writing is clear (even if pushing some general notions upper into the introduction would have been helpful)
The results and the methodology match the authors claims.

# Weaknesses
It takes a long time of reading before it is clear what is the method. Only in Sec 2.3 the authors state that the way in which an example is detected as memorised. Specifically, this is through SBERT similarity of the answer and question to all other questions and answers.

The paper uses the word memorisable, which, because they don't explain otherwise until the full method is explained, seems like a strong and promising trait of an example. Something along the lines of, a simple fact is likely to be memorised, while an answer that requires several deductions is much less likely to be.

What did you base your decisions on? I neither am able to understand what those numbers really represent, nor how you skipped from 81 to 60. " Since this approach is of course fallible we substracted a substantial number from this score to arrive at a memorisability threshold
of 60.0"

While TMLR stands to publish such work and this is not a reason to reject. I would state that my subjective judgment of the level of interest and novelty this work presents is low.

---

> ### Author Response · Authors · 2023-11-21
>
> Thank you for the good suggestions! We have incorporated them into a new revision of the paper and summarised below for convenience:
>
> #########
>
> Reviewer:  It takes a long time of reading before it is clear what is the method. Only in Sec 2.3 the authors state that the way in which an example is detected as memorised. Specifically, this is through SBERT similarity of the answer and question to all other questions and answers.
>
> Authors: The use of SBERT was originally mentioned in the Introduction, last paragraph before the contribution summary. We’ve now merged it into the second paragraph in the Introduction to make it more prominent along with the example you had suggested.
>
> #########
>
> Reviewer:  The paper uses the word memorisable, which, because they don't explain otherwise until the full method is explained, seems like a strong and promising trait of an example. Something along the lines of, a simple fact is likely to be memorised, while an answer that requires several deductions is much less likely to be.
>
>
> Authors: We’ve added an example into paragraph 2 of the Introduction along with merging the discussion about the use of SBERT.
>
> #########
>
> Reviewer:  What did you base your decisions on? I neither am able to understand what those numbers really represent, nor how you skipped from 81 to 60. " Since this approach is of course fallible we substracted a substantial number from this score to arrive at a memorisability threshold of 60.0"
>
> Authors: Thanks for pointing this out. We’ve rewritten this paragraph to provide specificity on the process we used to identify the memorisability threshold.
>
> #########
>
> Reviewer: While TMLR stands to publish such work and this is not a reason to reject. I would state that my subjective judgment of the level of interest and novelty this work presents is low.
>
> Authors: We respectfully offer our perspective that our paper offers a significant and interesting contribution along two dimensions:
>
> Firstly, we have noted (differing) issues with the experimental setup of prior approaches: (1) ability to control for pretraining data,  (2) needing to compare performance between different sets of “clean” and “dirty” samples, and/or (3) ability to detect discontinuous memorisable sequences. As far as we know the totality of our methods (semantic similarity with the “intervention” approach) is the only study on test-evaluation overlap to date that addresses all of these issues and we feel that this in itself is a significant contribution that others may find interesting.
>
> Secondly, we think our specific purpose of evaluating whether or not smaller models can “reason” versus memorise using our robust experimental setup is a “modestly significant” and topical contribution. Whilst at this point many might feel the issue is settled, at least for LLMs, the reality is that a number of respected figures continue to question the degree to which models can “reason” versus being stochastic parrots. E.g, https://www.nature.com/articles/d41586-023-02361-7 ChatGPT broke the Turing test — the race is on for new ways to assess AI. Nature, 619(7971):686–689.,   and https://www.nytimes.com/2023/03/08/opinion/noam-chomsky-chatgpt-ai.html .
>
> #########
>
> Reviewer:  Papers discussing the order of learning (e.g. The grammar-learning trajectories of neural language models‏, and Many of Guy Hacohen's non NLP works) often show that models learn in the same order, which might have a different prediction than more memorisation in larger ones, "just so" but memorisation when stopped not memorizing. This relates to noisy data learning, and to the long-tail hypothesis, which states memorization is itself learning See Feldman's works (e.g. What Neural Networks Memorize and Why: Discovering the Long Tail via Influence Estimation)
>
> Authors: Thank you for these suggestions. We’ve added discussion on them into the Related Work section. Please let us know if you think we have overlooked other relevant studies.
>
> #########
>
> Reviewer:  When you cite someone in line, use \citet{} (for example in the intro at e.g. (Carlini...) Should be e.g., Carlini (20...) or "For example (Chowdhery et al., 2022)" )
> Also consider using \citet[before;][after] or \citealp in cases of parenthesis just before\after the citations (e.g., Khashabi et al. (2020; 2022) (appendix B),).
>
> Authors: Thank you, we’ve fixed these.
>
> #########
>
> Reviewer:  Fig 1 is not easily interpretable + it has too many colors and styles, which instead of making the most important things pop, it makes everything compete on attention and just feel cluttered.
>
> Authors: We’ve removed the highlights on words and rounded the similarity scores. Hopefully this is now less cluttered. Let us know if you have any other suggestions for improving the diagram.
>
> #########
>
> Reviewer: Why 100 .0 and 2 .0?
> Authors: We’ve now made these integers thank you.

---

### Review · Reviewer_2MzY · 2023-11-03

**Summary Of Contributions:**

The main contribution of the paper is evaluating *unmemorisable* subset of reading comprehension dataset after training with additional synthetic training dataset (TDND)that are generated to teach numerical reasoning strategy.

I am not sure the contribution of this paper is big enough to justify publication at TMLR. The data augmentation is not their contribution, the only *new* thing is evaluating under new evaluation regime which are split by “memorized” and “unmemorized” evaluation set. However, such evaluation idea is not new, explored in prior work like Lewis et al.

**Audience:**

No

**Claims And Evidence:**

Yes

**Requested Changes:**

* Figure 1 is really not clear, what are each sample represent and each color represent? Things should be clearly labeled.

* Missed relevant work
Quantifying Train-Evaluation Overlap with Nearest Neighbors, FIndings of ACL 2023
https://aclanthology.org/2023.findings-acl.183.pdf

* Having the same hyperparameters (sec 2.1) — I’m not sure this is good, how are these hyper parameters chosen? Another reasonable approach would be to do grid search of hyper parameter for each setting.

**Strengths And Weaknesses:**

Strength: * The paper is written clearly, and experimental procedure is sound
* The experiments are conducted on a large range of benchmark dataset.


Weaknesses:
* The main experimental results are not novel. I am not sure what new insights these experimental results provide.
* Running experiments on other LM would be helpful.
* Similarity computation method can be studied a bit more carefully. Table 3 shows some examples, but quantitatively studying (on a random set of examples) where these sentence embedding can correctly measure similarity between examples would be helpful.

---

> ### Author Response · Authors · 2023-11-21
>
> Thank you for the good suggestions! We have incorporated them into a new revision of the paper and summarised below for convenience:
>
> #########
>
> Reviewer: I am not sure the contribution of this paper is big enough to justify publication at TMLR. The data augmentation is not their contribution, the only new thing is evaluating under new evaluation regime which are split by “memorized” and “unmemorized” evaluation set. However, such evaluation idea is not new, explored in prior work like Lewis et al.
> The main experimental results are not novel. I am not sure what new insights these experimental results provide.
>
> Authors: We respectfully suggest that our paper offers a significant and interesting contribution along two dimensions:
>
> Firstly, we have noted (differing) issues with the experimental setup of prior approaches: (1) ability to control for pretraining data,  (2) needing to compare performance between different sets of “clean” and “dirty” samples, and/or (3) ability to detect discontinuous memorisable sequences. As far as we know the totality of our methods (semantic similarity with the “intervention” approach) is the only study on test-evaluation overlap to date that addresses all of these issues and we feel that this in itself is a significant contribution that others may find interesting.
>
> Secondly, we think our specific purpose of evaluating whether or not smaller models can “reason” versus memorise using our robust experimental setup is a “modestly significant” and topical contribution. Whilst at this point many might feel the issue is settled, at least for LLMs, the reality is that a number of respected figures continue to question the degree to which models can “reason” versus being stochastic parrots. E.g, https://www.nature.com/articles/d41586-023-02361-7 Celeste Biever. 2023. ChatGPT broke the Turing test — the race is on for new ways to assess AI. Nature, 619(7971):686–689.,   and https://www.nytimes.com/2023/03/08/opinion/noam-chomsky-chatgpt-ai.html .
>
> #########
>
> Reviewer: Running experiments on other LM would be helpful.
>
> Authors: While we agree it would have been ideal, we hope that our decision to allocate our budget towards training three versions of each of the two BART settings with a view towards a robust evaluation of one model is sufficient mitigation.
>
> #########
>
> Reviewer: Similarity computation method can be studied a bit more carefully. Table 3 shows some examples, but quantitatively studying (on a random set of examples) where these sentence embedding can correctly measure similarity between examples would be helpful
>
> Authors: Noting that in addition to Table 3, Table 4 shows examples with Similarity Scores ranging from 77 to 95 and appendices D and E show more examples with scores around 59, we feel there are sufficient examples shown in the current version. If you still feel otherwise please let us know and we will add a table of random examples.
>
> #########
>
> Reviewer: Figure 1 is really not clear, what are each sample represent and each color represent? Things should be clearly labeled.
>
> Authors: We’ve removed the highlights on words and rounded the similarity scores. Hopefully this is now less cluttered. We also added horizontal dashed black lines to better indicate what the top text boxes are labelling. Please let us know if you have any other suggestions for improving the figure.
>
> #########
>
> Reviewer: Missed relevant work Quantifying Train-Evaluation Overlap with Nearest Neighbors, FIndings of ACL 2023 https://aclanthology.org/2023.findings-acl.183.pdf
>
> Authors: Thank you for this reference. We’ve added it into related work at the bottom of the first paragraph.
>
> #########
>
> Reviewer: Having the same hyperparameters (sec 2.1) — I’m not sure this is good, how are these hyper parameters chosen? Another reasonable approach would be to do grid search of hyper parameter for each setting.
>
> Authors: We’ve added a bit more verbiage into sec 2.1 overviewing our hyperparameter selection process and also noting that our best model selection method allows for some setting-specific optimization.

---

### Review · Reviewer_9xvj · 2023-11-13

**Summary Of Contributions:**

This paper focuses on the reading comprehension task, to study whether an existing dataset TDND – which proves to improve reading comprehension performance – enhances the results through memorization or generalization. Specifically, the authors identify the least similar or unmemorizable test instances to construct new test sets, finding that the TDND dataset can still improve the results on such test data, with decreased gains.

**Audience:**

Yes

**Claims And Evidence:**

No

**Requested Changes:**

I think there are significant changes to be made to turn this paper into a strong submission. For example, studying this problem for more tasks and with more model variants and insightful analysis. These changes would basically make this submission completely different. Within the scope of the current submission (only on the reading comprehension task with (a bit) out-of-dated models), it is difficult to convince me any trivial updates will change my opinion to recommendation.

**Strengths And Weaknesses:**

### Strengths:

1. The overall problem of focus is interesting, on the discussion of memorization or generalization to achieve the benchmark gains.
2. The designed experiments are reasonable.


### Weaknesses:

1. This paper only focuses on the reading comprehension task, the experiments are also only with one model, which is Bart that is not very representative these days. These settings are too limited to be considered good contributions.
2. The novel contents in this paper are not enough for a long paper with 9 pages – I feel the contents are more suitable for a 4-page submission.
3. The proposed methods and the pipelines are not novel, quite like a direct extension of existing work to the reading comprehension task – contributions are limited.
4. It seems the authors are trying to extend the definition of "memorization" by considering discontinuous tokens for example. However, it is difficult to prove that the memorization defined by the authors are complete, there might be other spurious relations which were not considered in this paper but lead to the remaining gains of TDND.

---

> ### Author Response · Authors · 2023-11-21
>
> Thank you for the review.  We respectfully suggest that our paper offers a significant and interesting contribution along two dimensions:
>
> Firstly, we have noted (differing) issues with the experimental setup of prior approaches: (1) ability to control for pretraining data,  (2) needing to compare performance between different sets of “clean” and “dirty” samples, and/or (3) ability to detect discontinuous memorisable sequences. As far as we know the totality of our methods (semantic similarity with the “intervention” approach) is the only study on train-evaluation overlap to date that addresses all of these issues and we feel that this in itself is a significant contribution that others may find interesting. We note that we have not claimed to have “proven” that no other spurious mechanism than memorisation could be responsible but suggest that the burden of proof would lie in at least suggesting what such an alternative mechanism could be and how that is different from memorisation.
>
> Secondly, we think our specific purpose of evaluating whether or not smaller models can “reason” versus memorise using our robust experimental setup is a “modestly significant” and topical contribution. Whilst at this point many might feel the issue is settled, at least for LLMs, the reality is that a number of respected figures continue to question the degree to which models can “reason” versus being stochastic parrots. E.g, https://www.nature.com/articles/d41586-023-02361-7 Celeste Biever. 2023. ChatGPT broke the Turing test — the race is on for new ways to assess AI. Nature, 619(7971):686–689.,   and https://www.nytimes.com/2023/03/08/opinion/noam-chomsky-chatgpt-ai.html .

---

### Decision · Action_Editor_SmJu · 2023-12-27

**Recommendation:** Reject

**Comment:**

The general question is one of interest to the community, and potentially the creation of data splits, but the experiments themselves are not. Without an updated and more comprehensive set of models it's not clear that the results are general or would impact the work others are doing within the broader NLP community.  Resubmission of a more comprehensive study would be welcome.

**Audience:**

This is the main concern.  All three reviewers felt that the results here do not generalize to settings (e.g. contemporary models) of interest to the community.  Note that the submission also doesn't try to explain the modeling choice/setting which leaves the audience question particularly hard to answer.  If small models are necessary for efficiency/power concerns BART is a strange choice, if the issue is open data for avoiding contamination there are better models in that domain as well, etc.  It's, therefore, hard to justify the choices made here or what research would be informed by the experiments.

**Claims And Evidence:**

The authors aim to study the bounds of memorization and generalization within language models (a particular focus on smaller models).  This involves creating (non-)memorization data splits and investigating how embedding similarity correlates when looking at two evaluation datasets.  They have a set of unseen datasets (commonly used in the literature) and based their evaluation on training of BART models.

**Resubmission Of Major Revision:**

The authors may consider submitting a major revision at a later time.